# Real-time precision opto-control of chemical processes in live cells

Matthew G. Clark[1], Gil A. Gonzalez[1], Yiyang Luo[1], Jesus A. Aldana-Mendoza[1], Mark S. Carlsen[1], Gregory Eakins [1], Mingji Dai [1,2] & Chi Zhang [1,2,3] ✉

Precision control of molecular activities and chemical reactions in live cells is a long-sought capability by life scientists. No existing technology can probe molecular targets in cells and simultaneously control the activities of only these targets at high spatial precision. We develop a real-time precision opto-control (RPOC) technology that detects a chemical-specific optical response from molecular targets during laser scanning and uses the optical signal to couple a separate laser to only interact with these molecules without affecting other sample locations. We demonstrate precision control of molecular states of a photochromic molecule in different regions of the cells. We also synthesize a photoswitchable compound and use it with RPOC to achieve site-specific inhibition of microtubule polymerization and control of organelle dynamics in live cells. RPOC can automatically detect and control biomolecular activities and chemical processes in dynamic living samples with submicron spatial accuracy, fast response time, and high chemical specificity.

In biological science, there is a critical need to control molecular activities at high spatial precision. The capability to activate drugs at only selected locations can help better understand spatial distributions of targets and rule out off-target effects; controlling enzyme activities at micrometer spatial precision and only in specific organelles allows better elucidation of their roles in cellular activities; manipulating single organelles and control their behaviors without affecting other parts of the cell shed new light on their functions. No biotechnologies are available to achieve such chemical-selective precision control. The conventional chemical treatment by culturing cells with compounds has poor spatial delivery selectivity and might pose off-target effects. Optical tweezers and trapping can only physically manipulate a few pre-detected targets[1–3]. Current laser ablation and manipulation methods are based on pre-image acquisition and manual operation of laser beams to interact with the target of interest[4–7]. Optogenetics methods can control functions of neurons using light radiation and light-sensitive ion channels, however, require pre-imaging and demonstrate little subcellular precision[8–10]. Live biological samples are highly dynamic with fast-moving intercellular organelles and active migration processes. Therefore, existing optical

manipulation technologies that are based on pre-imaging and analysis do not apply to these dynamic systems. A suitable technology requires simultaneous detection and control of selected molecular targets in real-time with high chemical selectivity and spatial accuracy.

In this work, we develop a real-time precision opto-control (RPOC) technology that can detect and control molecules simultaneously, selectively, and precisely at only desired activity sites. First, during laser scanning, an optical signal is generated at a specific pixel from target molecules. Then, the detected optical signal is compared with preset values using comparator circuitry. A desired optical signal will activate an acousto-optic modulator (AOM) which is used as a fast switch to couple another laser beam to interact at the same pixel. The optical signal detection, processing, and opto-control happen in real-time during laser scanning. Digital logic functions allow opto-control of molecular activities based on the logic output from multiple signal channels. RPOC can accurately detect and control biomolecules in real-time without affecting other locations of the sample. It is highly chemically selective since the optical signal can be selected from a range of responses such as fluorescence and Raman. Using a photochromic compound, we showed precision control of the state of these

[1]Department of Chemistry, Purdue University, 560 Oval Dr., West Lafayette, IN 47907, USA. [2]Purdue Center for Cancer Research, 201 S. University St., West Lafayette, IN 47907, USA. [3]Purdue Institute of Inflammation, Immunology, and Infectious Disease, 207 S. Martin Jischke Dr., West Lafayette, IN 47907, USA. ✉e-mail: zhan2017@purdue.edu

molecules selectively in different regions of the cells. We also synthesized a photoswitchable microtubule polymerization inhibitor and deployed RPOC to enable site-specific precision control of microtubule polymerization and manipulation of lipid droplet trafficking in live cells. RPOC offers an exciting way to automatically and selectively control molecular activities and chemical reactions by photoactivatable molecules with submicron spatial precision during imaging.

## Results

### The RPOC platform

The concept of RPOC, which is based on fast laser scanning, is illustrated in Fig. 1a, b. A laser(s) for optical signal excitation is scanning through the field of view. During the laser scanning, if a chemical-specific optical signal is detected and satisfies a preset condition (e.g., surpasses a threshold value), it will trigger an AOM to send a separate laser beam to interact with the sample at the same pixel in real-time. Optical signals that do not satisfy this condition will "turn off" the control laser beam to avoid laser interaction. Digital comparator circuits (Supplementary Fig. 1, Note 1) are designed for presetting the

selection conditions (e.g., the threshold $V_T$), performing analog/digital comparisons, and sending out a standard transistor-transistor logic (TTL) voltage for AOM control. A schematic of the RPOC system is shown in Fig. 1c. A dual-output femtosecond laser is used to perform optical signal excitation and opto-control. The 1045 nm laser output is used as the Stokes beam and the frequency-tunable laser output is used as the pump beam for stimulated Raman scattering (SRS) signal generation[11–13]. The laser pulses are also used for two-photon excitation fluorescence (TPEF) signal excitation[14]. Portions of both outputs are frequency-doubled to the visible range for opto-control. The selected control laser is sent to an AOM and commanded by the comparator circuits. The laser beam profiles after the AOM using '0' and '1' TTL commands are shown in Fig. 1d, e, respectively. The 0th order output is blocked by a beam stop and the 1st order of the AOM output is combined with the excitation IR laser beams by a dichroic beam splitter before coupling into the microscope. Figure 1f shows the ~15 ns response time of the comparator circuits. The AOM response time (Supplementary Note 2) is calculated to be ~7 ns. Considering the optical path difference and cable length, the response time of the RPOC system using fluorescence signals is <50 ns (Supplementary

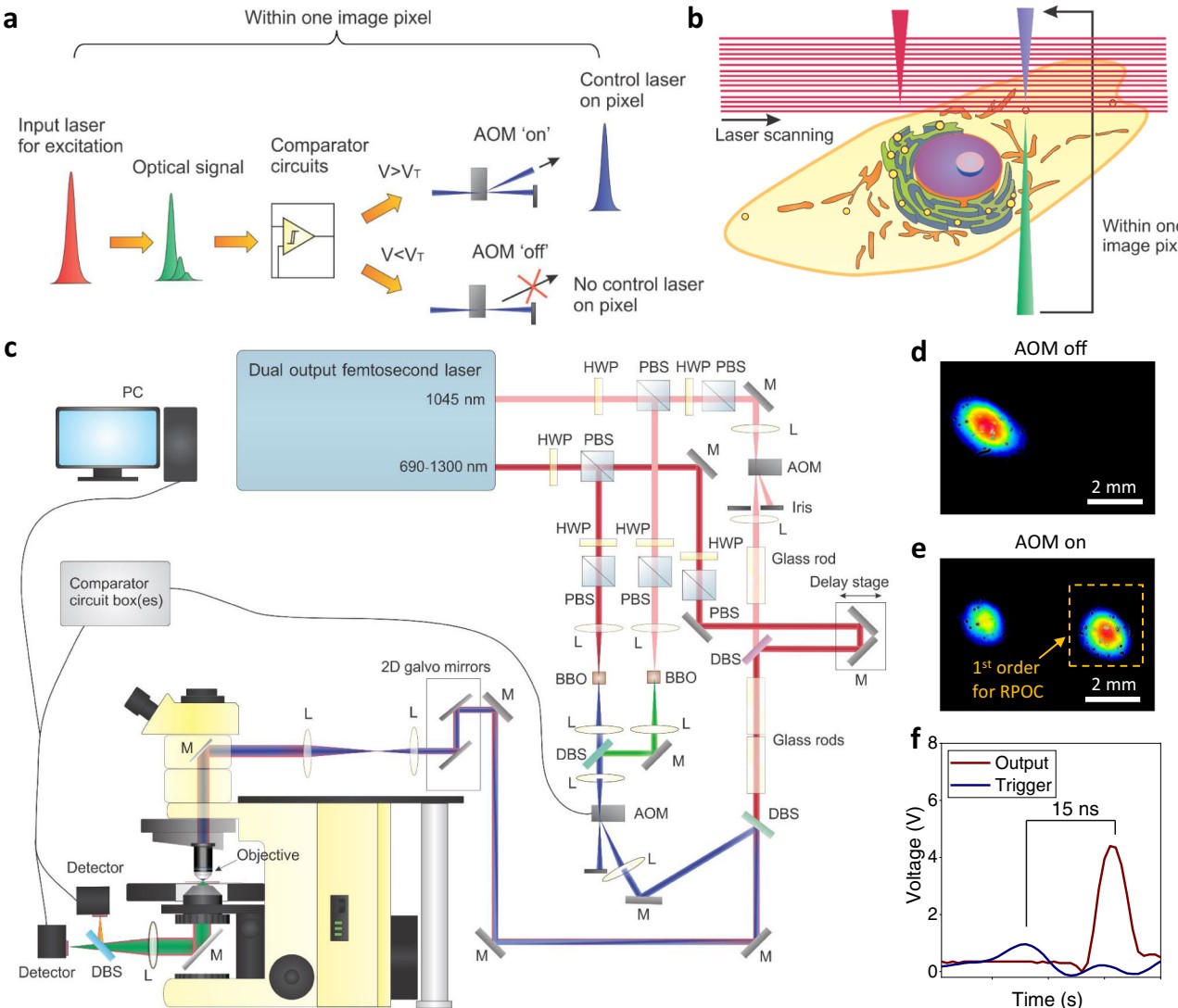

**Fig. 1 | The RPOC concept and optical configuration. a** An illustration of the RPOC concept. **b** An illustration of RPOC for selective control of molecular activities in a cell during laser scanning. **c** A schematic of the RPOC experimental setup. HWP half-wave plate, L lens, M mirror, DBS dichroic beam splitter, PBS polarization beam splitter, AOM acousto-optic modulator, BBO Beta Barium Borate. **d** The profile of

the control laser beam (at 522 nm) after the AOM when the AOM is turned off. **e** The profile of the control laser beam after the AOM when the AOM is turned on (the 1st order deflection is highlighted). **f** The response time of the comparator box is measured to be ~15 ns.

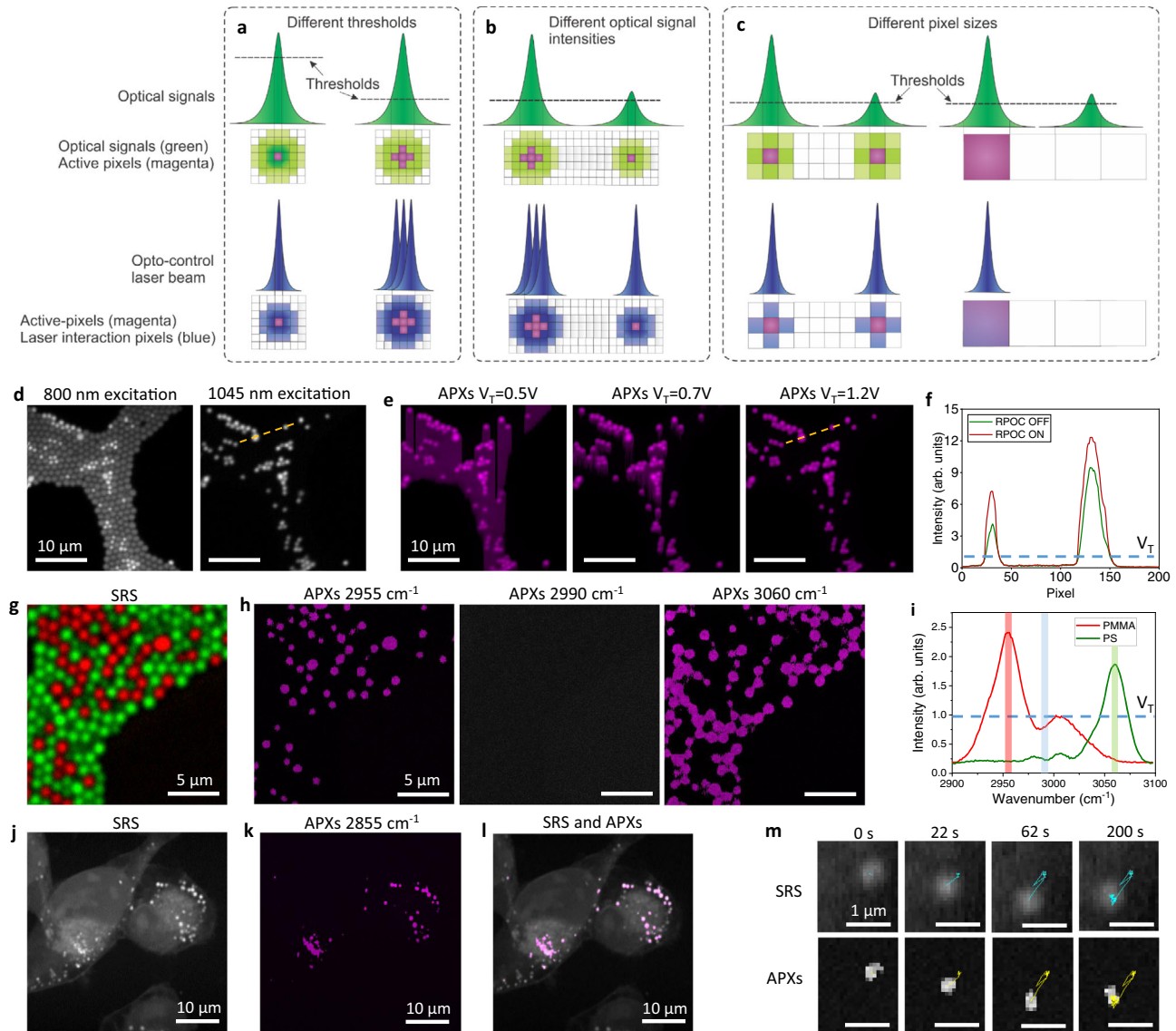

**Fig. 2 | Mapping out RPOC active pixels (APXs). a** An illustration of APX selection using different signal thresholds in the over-sampling condition. The green pulses and pixels indicate optical signals, the magenta pixels indicate APXs, and the blue pulses and pixels indicate the interaction pulses and pixels. **b** An illustration of APX selection using different optical intensities in the over-sampling condition. **c** An illustration of APX selection at larger pixel sizes for different optical signals. **d** Mixed fluorescent particles detected in the 570/60 nm channel excited by 800 nm pulses (left) and 1045 nm pulses (right). **e** APXs determined using different voltage threshold $V_T$ values for the signals from the 570/60 nm channel. **f** Comparing the 570/60 nm channel optical intensity when the RPOC is turned on (red) and off (green). **g** A pseudo-color stimulated Raman scattering (SRS) image containing PMMA (red) and PS (green) particles. **h** APXs determined using the PS peak at 2955 cm$^{-1}$ (left), PMMA peak at 3060 cm$^{-1}$ (right), and no Raman peaks at 2990 cm$^{-1}$ (middle). **i** SRS spectra of PMMA (red) and PS (green). The red, green, and blue bars indicate wavenumbers used for RPOC in panel **h**. **j** An SRS image of MIA PaCa-2 cells in the lipid CH$_2$ stretching region. **k** APXs determined using SRS signals from lipid droplets (LDs). **l** An overlay of the SRS image and the APXs turned on only at the LDs. **m** Time-lapse SRS images of an LD in a live MIA PaCa-2 cell (top row) and the corresponding APXs (bottom row) determined by the SRS signals. The color curves plot trajectories of the LD and the APXs in 200 s.

Note 3), much shorter than the 10 μs pixel dwell time for laser scanning. The response time of RPOC using SRS signals is majorly determined by the lock-in time constant, which might affect no more than one additional pixel which corresponds to ~90 nm in our oversampling condition. The spatial resolution of the SRS and TPEF modalities is measured to be 373 nm (Supplementary Fig. 2 and Supplementary Note 4). The RPOC control laser beam gives a spatial precision of 525 nm (Supplementary Fig. 3 and Supplementary Note 5).

### Real-time control of active pixels (APXs) using chemical-specific optical signals

An active pixel (APX) is defined as the pixel location at which the control laser beam is turned on. Tracking APXs is critical for visualizing the opto-control locations. In the over-sampling condition (the pixel size is smaller than the laser beam size at the focus), the size of the laser interaction area is larger than the size of the APXs (Fig. 2a). Higher intensity thresholds reduce APXs and laser interacting areas. Similarly, at the same intensity threshold, weaker optical signals above the threshold reduce APXs and the laser interaction areas (Fig. 2b). Increasing the pixel size might change these properties. Different optical signal intensities that would result in different APXs in the oversampling condition might give the same APXs and interaction areas (Fig. 2c). When the pixel size is greater than the actual laser beam size, the APXs might be the same or even larger than the actual interaction area (Fig. 2c). The oversampling condition is used throughout this work.

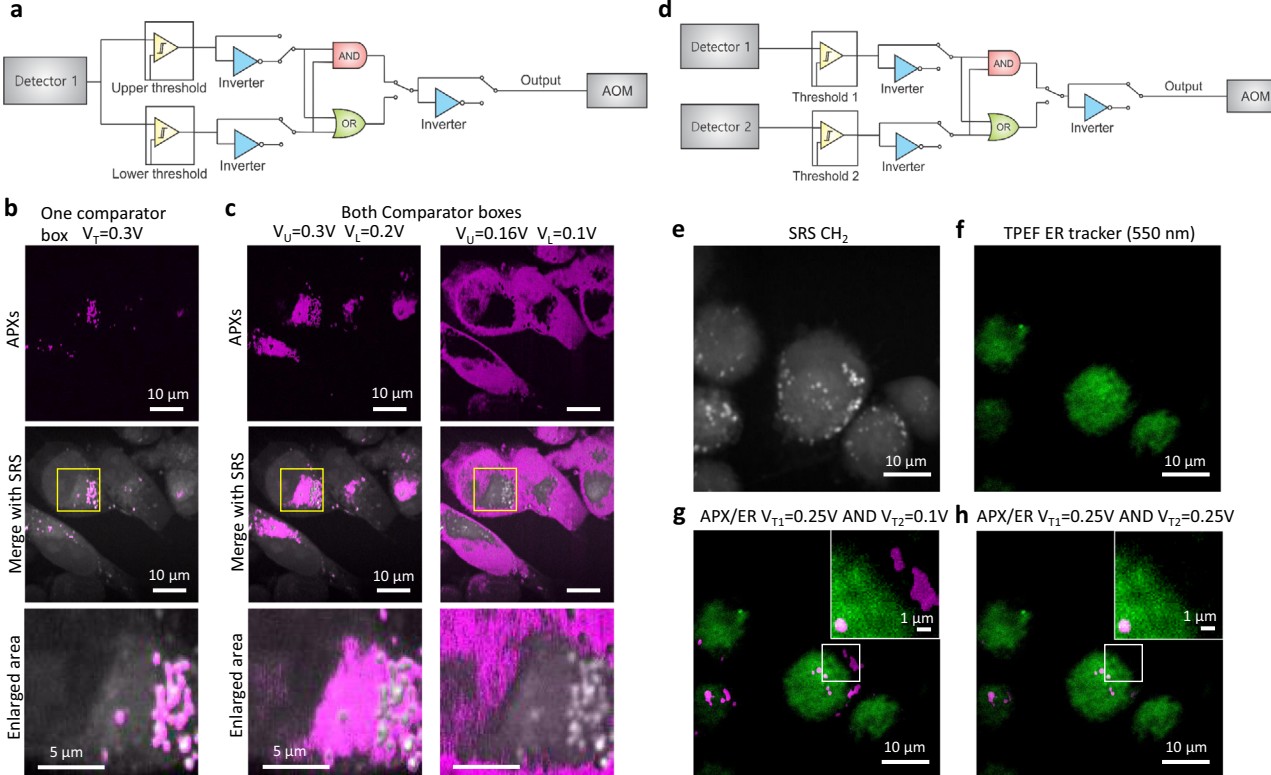

**Fig. 3 | Digital logic control of active pixels (APXs). a** An illustration of electronics to select an intensity passband for the determination of APXs. **b** The APXs selected using only one comparator box with $V_T = 0.3$ V (upper panel), the overlay of the APXs with the corresponding SRS image from the same field of view (middle panel), and a magnified image from the selected area (bottom panel). **c** APXs selected using two comparator boxes with the intensity range between 0.2–0.3 V (left panels) and 0.1–0.16 V (right panels). **d** Illustration of electronics to choose the AND function for determination of APXs using two comparator boxes. **e** An stimulated Raman scattering (SRS) image of MIA PaCa-2 cells at the $CH_2$ stretching vibration. **f** A two-photon excitation fluorescence (TPEF) image of the MIA PaCa-2 cells labeled using endoplasmic reticulum (ER) Tracker. **g** An overlay of the TPEF image from the ER and the APXs determined using $V_{T1} = 0.25$ V (SRS) and $V_{T2} = 0.1$ V (TPEF). **h** An overlay of the TPEF image from ER and the APXs using $V_{T1} = 0.25$ V (SRS) and $V_{T2} = 0.25$ V (TPEF).

First, we demonstrate the importance of the threshold voltage for selecting APXs. We use fluorescence signals from mixed orange and green microparticles to demonstrate how to determine APXs. Figure 2d shows fluorescence signals detected in the 570/60 nm fluorescence channel excited by 800 nm (left panel) and 1045 nm (right panel) picosecond laser pulses. The latter only shows the orange particles and is paired with a comparator circuit box to determine the APXs. The selection condition is the fluorescence signal above a preset threshold $V_T$. When the $V_T$ is low (0.5 V and 0.7 V), APXs exceed the fluorescence pixels in the 570/60 nm channel (Fig. 2e). At a proper threshold $V_T = 1.2$ V, APXs perfectly overlay with fluorescence signals. Figure 2f compares the intensity profiles at selected lines in Fig. 2d, e when RPOC is turned off and on. The intensity increase is contributed by the leaking of the control laser which is only turned on by the fluorescence signal at the particles.

Next, we demonstrate an example of APX detection using Raman signals. Figure 2g shows chemical maps of mixed poly(methyl methacrylate) (PMMA) and polystyrene (PS) particles generated by hyperspectral SRS microscopy[15]. Using Raman shifts at 2955 cm$^{-1}$ or 3060 cm$^{-1}$, we can determine APXs using either PMMA or PS SRS signals (Fig. 2h). The selection condition is optical signals >$V_T = 1$ V. Figure 2i displays the SRS spectra of PMMA and PS, the $V_T$, and the selected Raman shifts for APX determination in Fig. 2h. APXs can be selected on different chemicals in real-time by tuning laser frequencies to match different Raman transitions during imaging (Supplementary Movie 1).

Finally, to demonstrate the real-time APX determination capability, we track the lipid droplet (LD) mobility in MIA PaCa-2 cells and

their respective selected APXs to confirm similar trajectories. We use the lipid $CH_2$ symmetric stretching SRS signals at 2855 cm$^{-1}$ for LD imaging and we can automatically select APXs only at the LDs in live cells, as shown in Fig. 2j–l. Side-by-side time-lapse images of SRS and APX selection using the lipid signals are shown in Supplementary Movie S2. The trajectory of APXs triggered by a single LD matches the corresponding LD trajectory (Fig. 2m, Supplementary Movie 3). We also demonstrate 3D precision controlling of the APXs in MIA PaCa-2 cells, as shown in Supplementary Movies 4, 5. These results highlight the capability of tracing intercellular dynamics for APX determination using RPOC.

## Digital logic control of APXs

A second comparator box with digital logic functions is also designed as shown in Supplementary Fig. 4, Note 6. Digital logic functions can be selected from AND, OR, NAND, and NOR. Using the two comparator circuit boxes, we can select any intensity range from a single detector for APX determination. The connections to achieve this function are illustrated in Fig. 3a and Supplementary Fig. 5. Figure 3b shows APXs selected on the LDs using only comparator box 1 and a single intensity threshold. Figure 3c shows APXs determined using different intensity passbands between the upper and lower thresholds. The APXs selected between $V_T = 0.2$–0.3 V are more associated with the endoplasmic reticulum (ER) and between 0.1–0.16 V are mostly on cytosol and nuclei. Spectral phasor analyses of hyperspectral SRS images of the same cells, which segment different cellular compartments[16], are shown in Supplementary Fig. 6 and Supplementary Note 7 for comparison.

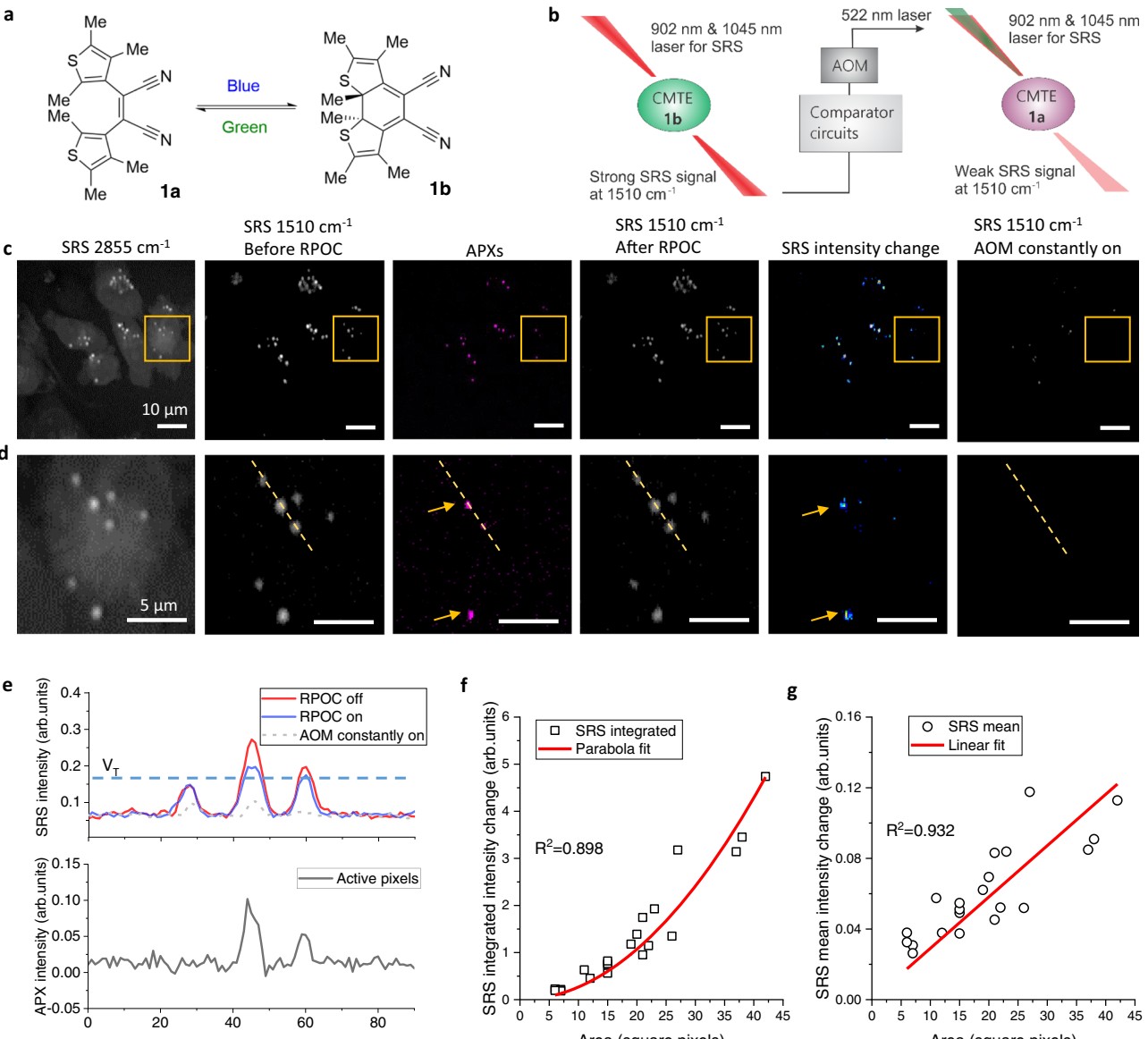

**Fig. 4 | Precision control and quantitative comparison of site-specific chemical changes using RPOC. a** An illustration of switching states of CMTE between the open *cis* isomer (**1a**) and the closed isomer (**1b**) forms using blue and green light. **b** An illustration of the workflow for CMTE conversion by the RPOC using the 1510 cm⁻¹ SRS signal and a 522 nm laser. AOM: acousto-optic modulator. **c** An stimulated Raman scattering (SRS) image at 2855 cm⁻¹, at 1510 cm⁻¹ before RPOC, the active pixels (APXs), at 1510 cm⁻¹ after RPOC using $V_T = 0.17$ V, the SRS intensity difference before and after RPOC, at 1510 cm⁻¹ after the AOM constantly on for 20

frames. **d** Magnified images of the selected areas in panel **c. e** SRS intensity profiles of images and APXs along the dotted lines in panel **d.** The dashed curve shows the SRS intensity profile along the same line after 20 scan frames with AOM constantly on. **f** Integrated SRS intensity changes of CMTE as a function of the number of APXs for CMTE aggregates. Open squares are experimental results, the curve is the quadratic fitting. **g** Mean SRS intensity changes of CMTE as a function of the number of APXs for CMTE aggregates. Open circles are experimental results, the line is the linear fitting.

The connections for implementing the digital logic functions using two comparator boxes and two detectors are illustrated in Supplementary Fig. 7. The selection of the AND function is illustrated in Fig. 3d. We first demonstrate the AND function using mixed fluorescent PS particles, nonfluorescent PS particles, and nicotinamide adenine dinucleotide hydrogen (NADH) crystals, as shown in Supplementary Figs. 8,9, and Note 8. The AND function allows determining APXs only on the fluorescent PS particles that show up in both TPEF and SRS channels. Next, we used SRS to excite lipid signals (Fig. 3e) in MIA PaCa-2 cells and label the cells using a fluorescent ER-Tracker which can be visualized in the TPEF channel (Fig. 3f). By using an appropriate $V_{T1}$ in the SRS channel and a low $V_{T2}$ in the TPEF channel, APXs can be selected from most of the LDs in the cells (Fig. 3g). Increasing the $V_{T2}$ in the TPEF channel can exclude the LDs outside the

ER and select APXs on LDs only on the ER (Fig. 3h). These results demonstrate using the AND logic from two separate detectors for APX determination. The connections of OR, NAND, and NOR functions for RPOC are illustrated in Supplementary Fig. 10.

**Controlling photochromic molecules at submicron precision**

To demonstrate precision control of chemical processes using the RPOC, we used a photochromic molecule, *cis*−1,2-dicyano1,2-bis(2,4,5-trimethyl-3-thienyl)ethene (CMTE), which can be changed from its open *cis* isomer (**1a**) to closed isomer (**1b**) by UV light and switched back by visible light at 520 nm (Fig. 4a)[17,18]. A strong Raman signature peak at 1510 cm⁻¹ can be detected for **1b** but not **1a**[17]. The SRS signal from this peak can be visualized by tuning the pump beam to 902 nm. We found that the combination of the pump and Stokes pulsed lasers

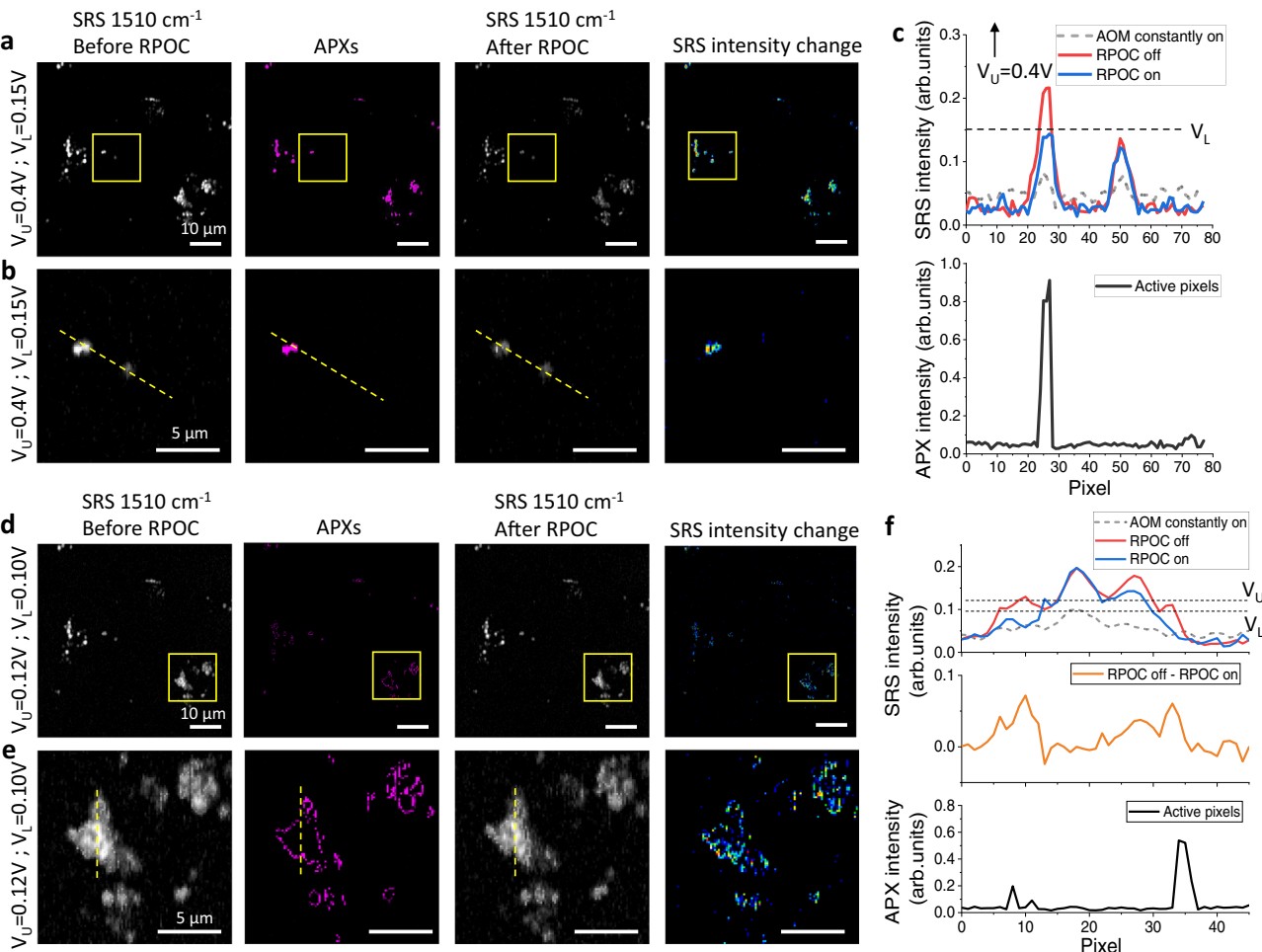

**Fig. 5 | Precision control of chemical changes by RPOC using different optical intensity ranges. a** A stimulated Raman scattering (SRS) image of CMTE at 1510 cm⁻¹ before RPOC (left), APXs selected by an upper threshold ($V_U$) of 0.4 V and lower threshold ($V_L$) of 0.15 V (middle left), an SRS image at 1510 cm⁻¹ after the RPOC of the CMTE (middle right), and the SRS intensity difference before and after RPOC (right). **b** Magnified images from the highlighted regions in panel **a**. **c** SRS and APXs intensity profiles along the dotted lines in panel **b**. The dashed curve shows the SRS intensity profile along the same line after 20 scan frames with the acousto-optic modulator (AOM) constantly on. **d** Similar images as in panel **a**, using $V_U$ = 0.12 V and $V_L$ = 0.10 V for APX selection. **e** Magnified images from the selected regions in panel **d**. **f** SRS and APXs intensity profiles along the dotted lines in panel **e**. The dashed curve shows the SRS intensity profile along the same line after 20 scan frames with the AOM constantly on. The middle panel plots SRS intensity difference before and after RPOC.

for SRS imaging can also transform CMTE to **1b**. A 522 nm laser beam frequency-doubled from the 1045 nm laser output is used for RPOC to convert **1b** to **1a** at selected subcellular locations. The Raman transition at 1510 cm⁻¹ is used as the reporter for RPOC-induced **1b** to **1a** conversion during laser scanning (Fig. 4b).

First, we cultured MIA PaCa-2 cells with CMTE and observed an accumulation of CMTE in LDs of the cells due to the hydrophobic structure of the chemical (Fig. 4c, d). Then, we used RPOC to convert **1b** to **1a** at selected locations of the sample with a control laser beam power of ~10 μW. A single comparator box with a selection condition of $V_T > 0.17$ V was used to determine APXs which are majorly contributed by high-intensity CMTE aggregates, as shown in Fig. 4c, d. After five frames of RPOC laser scanning, SRS signals at 1510 cm⁻¹ from the APX-associated pixels are reduced, which can be visualized from the SRS intensity change image in Fig. 4c, d. The pixels where chemical conversion happened, as shown in the SRS intensity change images, agree with the APXs. If the control laser beam is constantly turned-on during laser scanning for 20 frames, SRS signals at 1510 cm⁻¹ on the entire image are significantly reduced (Fig. 4c, d). Figure 4e plots SRS intensity profiles along the dashed lines in Fig. 4d, for images before and after the RPOC, the APXs, and after nonselective laser control after 20 frames. We see that laser-induced chemical changes of CMTE only

happen at the APXs. Such chemical changes can be quantified by integrating the SRS intensity change of CMTE on APXs of each aggregate, which shows a quadratic dependence on the number of APXs from each aggregate (Fig. 4f). This nonlinear dependence arises from the oversampling condition used for RPOC, as illustrated in Fig. 2a. The mean SRS intensity change, on the other hand, has a near-linear dependence on the number of APXs (Fig. 4g). These analyses show that RPOC can not only selectively control chemical changes in space but also potentially quantify the amount of products and reaction rates.

RPOC can control CMTE at different parts of cells using various selection conditions. We first connected two comparator boxes as illustrated in Supplementary Fig. 5 to select an SRS signal range between two intensity levels. Here, the RPOC is only applied for a singleframe laser scanning with 10 μs dwell time per pixel. Figure 5a, b display the SRS signals from CMTE at 1510 cm⁻¹ before the RPOC, the APXs, after a single-frame RPOC, and the SRS intensity changes with a selection condition of $V_L$ = 0.15 V and $V_U$ = 0.4 V, where $V_L$ and $V_U$ are the lower and upper signal limits for RPOC, respectively. The SRS image of CH₂ stretching is shown in Supplementary Fig. 11. Since $V_U$ is very high, this selection range determines the APXs mostly from the centers of the aggregates that contribute to strong CMTE SRS signals (Fig. 5c), similar to using a single comparator box as shown in Fig. 4.

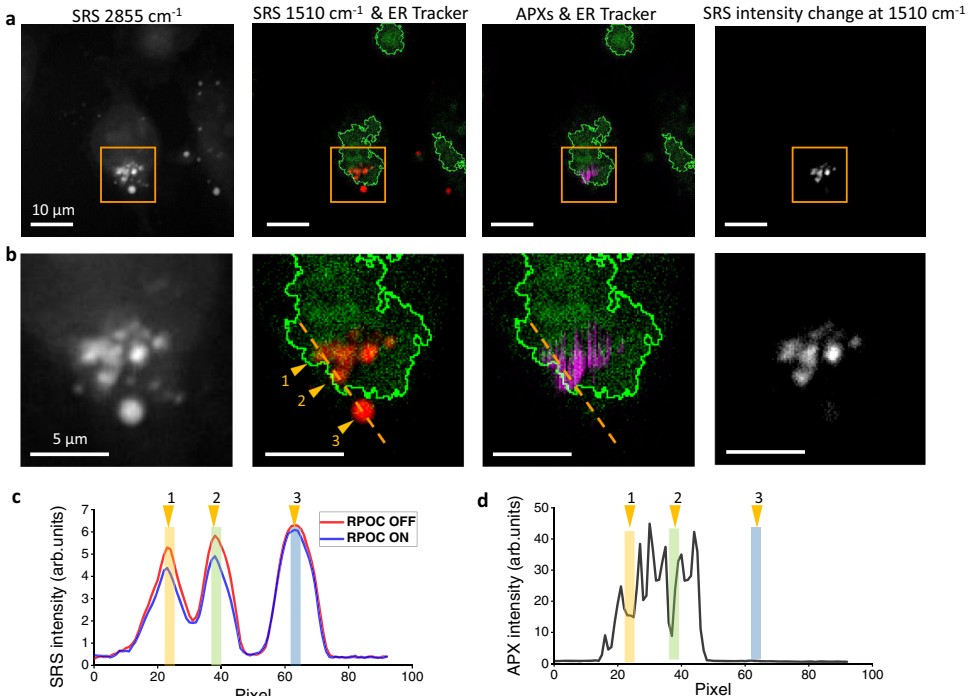

**Fig. 6 | Precision control of chemical changes by RPOC using digital logic from two detectors. a** Images of MIA PaCa-2 cells showing stimulated Raman scattering (SRS) signals at 2855 cm⁻¹ CH₂ stretching, the overlay of SRS signals from CMTE at 1510 cm⁻¹ (red) and endoplasmic reticulum (ER) Tracker fluorescence signals (green), the overlay of active pixels (APXs, magenta) and ER-Tracker signals (green), and the CMTE SRS signal changes after RPOC. **b** Magnified images from the highlighted areas in panel **a**. **c** CMTE SRS intensity profiles before and after RPOC along the selected lines in panel **b**. **d** Intensity profiles of APXs. Colored bars highlight the positions of selected CMTE aggregates in panel **b**.

When the selection condition is chosen as $V_L = 0.10$ V and $V_U = 0.12$ V, as shown in Fig. 5d, e, weaker SRS signals selected APXs from the edges of most aggregates. In this case, the centers of the aggregates, which are not associated with APXs, are left unchanged, while the molecules at the edges of the aggregates are converted from **1b** to **1a** by RPOC. Figure 5f plots the SRS intensity change and APXs along the dotted line and the intensity thresholds. The CMTE transition from **1b** to **1a** occurs only at the edges of the aggregates. There are pixels with noticeable SRS signal decrease outside APXs (Fig. 5f). This is due to the use of an oversampling condition in which the RPOC laser interacting range is larger than the APXs, as illustrated in Fig. 2a.

Next, we used RPOC to selectively control the **1b** to **1a** conversion accumulated only in ER-associated aggregates. SRS was used to detect CMTE targeting the 1510 cm⁻¹ peak while ER-Tracker in the TPEF (540-600 nm) channel was used to delineate ER in live MIA PaCa-2 cells. As shown in Fig. 6a, b, APXs can be determined on ER-associated aggregates using SRS and TPEF signals from two detectors with the digital AND function and appropriate threshold levels. The ER boundary is shown in Fig. 6a, b. By plotting the SRS intensity profiles of three aggregates (Fig. 6c) and their corresponding APXs (Fig. 6d) along the dashed lines (Fig. 6b), we find that CMTE in aggregate #3, which is not on the ER, is unaffected by RPOC; while CMTE in aggregates #1 and #2, both on the ER, are converted from **1b** to **1a** by RPOC. These results demonstrate that RPOC with digital logic functions can control molecular activities and chemical reactions associated with multiple organelles or related to organelle interactions.

### Precision control of microtubule dynamics and lipid droplet trafficking in live cells

To precisely control chemical activities in biological samples, photo-switchable inhibitors can be developed and used with RPOC. We synthesized a photoswitchable photostatin, a light-convertible inhibitor for microtubule polymerization, to control microtubule dynamics[19].

The *trans-* and *cis-* isomers of the photostatin PST-1 are shown in Fig. 7a. The *cis* form is active and can be converted from the inactive *trans* form using blue light. Green light, on the other hand, can reverse the process. Procedures for the synthesis of PST-1 are detailed in Supplementary Note 9. UV-Vis spectra of PST-1 after being illuminated with 405 and 532 nm light sources are shown in Fig. 7b, which matches the results from the previous publication[19]. To visualize the microtubule dynamics, a HeLa Kyoto cell line with stable EGFP co-expression with EB3, a protein that binds to the plus-end of microtubules, is used. We used TPEF to excite the EGFP-EB3 in cells. To increase the signal levels of EGFP-EB3, we bypassed the chirping rods in the optical beam path and used femtosecond lasers for excitation. We also averaged the images to enhance the visualization of the EB3 dynamics in cells. The cells are treated with 4 μM *trans*-PST-1 for 15 min before RPOC. Blue laser pulses at 400 nm frequency-doubled from 800 nm laser pulses are applied for RPOC. As shown in Fig. 7c left panel, before blue light interaction, the high-intensity streaks are averaged EB3 signals, showing their dynamic activities in HeLa cells. These features are mostly removed by scanning the blue light throughout the entire field of view (Fig. 7c, right panel), indicating the disruption of microtubule polymerization similar to the previous report[19].

The EGFP-EB3 proteins have weak TPEF signals and experience gradual photobleaching by the femtosecond excitation lasers. To visualize the EB3 activities, we designed the image acquisition processes as illustrated in Fig. 7d. Ten frames of TPEF images were acquired before RPOC for average (T1, red), followed by 5 frames of RPOC treatment (T2, magenta), 10 frames of TPEF images after RPOC (T3, green), and 20 frames of TPEF images after T3 (T4, red). Image frames in each time window are averaged to enhance the contrasts for comparison. Figure 7e shows pseudo-color overlapped images of T1 vs T3. Figure 7f displays APXs determined using SRS channels. The overlapped images of T3 vs T4 and with APXs are displayed in Supplementary Fig. 12. To visualize details, we enlarged two areas of the

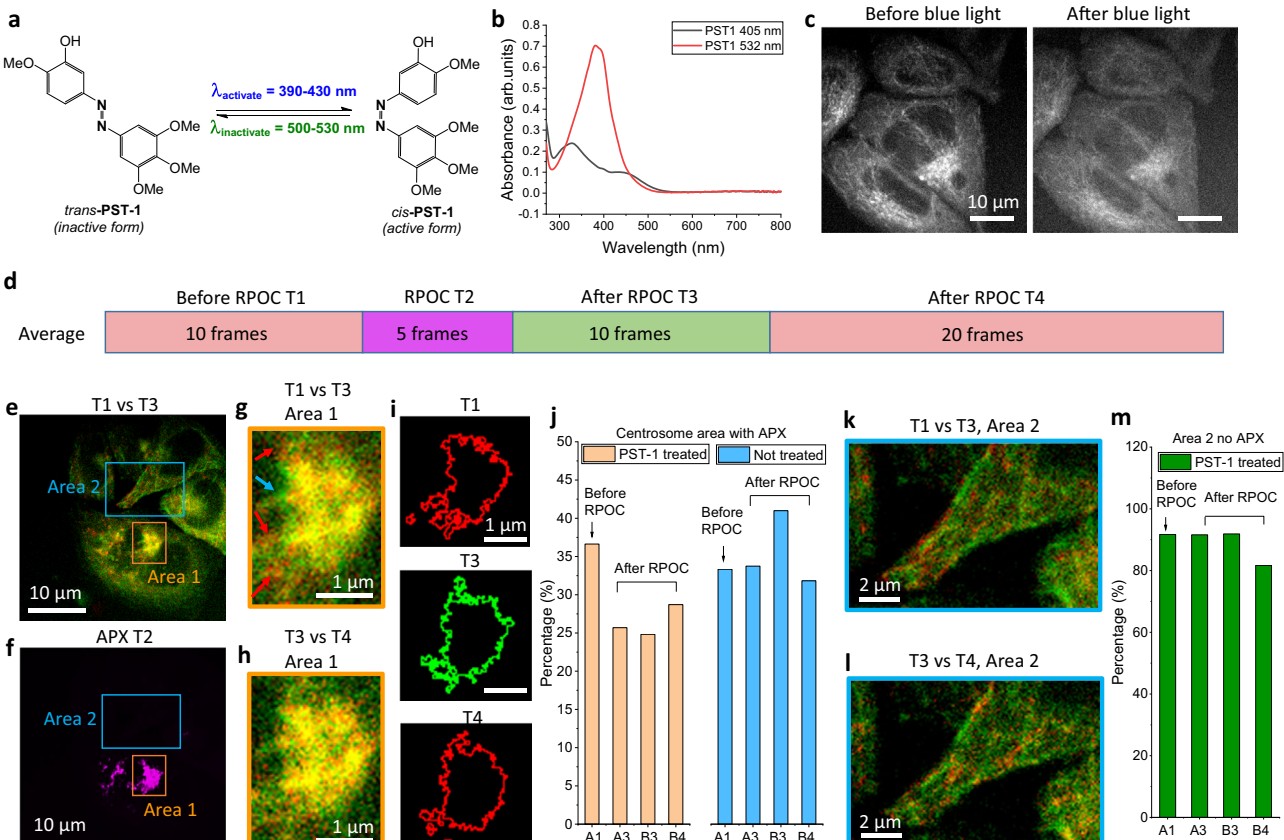

**Fig. 7 | Control of microtubule polymerization in cells using RPOC. a** Structures and light responses of PST-1. **b** UV-Vis spectra of PST-1 after blue and green light exposure. **c** Averaged two-photon excitation fluorescence (TPEF) images of EGFP-EB3 signals from transfected Kyoto HeLa cells before and after 400 nm light exposure. **d** Image analysis procedures and four time-windows for EGFP-EB3 TPEF signal comparison. **e** Overlays of EGFP-EB3 TPEF signals averaged for T1 (red) and T3 (green). **f** Active pixels (APXs) averaged in T2. Cells are treated with 4 µM PST-1 for 15 min before imaging and RPOC. **g**, **h** Area 1 selected in panel **e** (T1 vs T3) and

image selected in Fig. 7e and Supplementary Fig. 12a. Strong TPEF signals in Area 1 detect the centrosome locations in cells. We found that in the APX area (close to centrosomes), T1 vs T3 averages show strong signal disparities, as highlighted by arrows, indicating variation of EB3 dynamics before and after RPOC (Fig. 7g). However, after RPOC, as shown in Fig. 7h, T3 vs T4 show overlapping signals, indicating the stopped microtubule dynamics in this area. For better comparison, see movie S6. Intensity threshold outlines of TPEF signals for T1, T3, and T4 are shown in Fig. 7i. Using intensity thresholding, we quantified the signal changes by comparing T1, T3, and T4 images. The detailed method is explained in Supplementary Note 10. The results show that the TPEF image before RPOC at T1 has much more differences than those after RPOC at T3 and T4 (Fig. 7j). We performed a similar measurement by whole image blue light scanning for the cells without PST-1 treatment. The images showed significant signal differences for both T1 vs T3 and T3 vs T4 (Supplementary Figs. 14 and 15, Supplementary Movies 8 and 9), which were quantified in Fig. 7j. The untreated cells at all the time windows show similarly high-level differences to A1 in the PST-1 treated group, indicating unstopped microtubule dynamics in the centrosome areas. On the contrary, Area 2 is not associated with APXs, and both the T1 vs T3 and T3 vs T4 images show disparate contrast and features (Fig. 7k, l, Supplementary Movie 7). This area has a much higher signal difference (Fig. 7m) due to the absence of the centrosomes that contribute to a large portion of unchanged TPEF signals after blue light treatment.

Supplementary Fig. 12a (T3 vs T4), respectively. Arrows point out EGFP signal mismatches, indicating changing EB3 dynamics at the two time-windows. **i** TPEF intensity outlines of T1, T3, and T4 for Area 1. **j** Calculated percentage of TPEF signal changes for the three time-windows before and after RPOC in the centrosome areas for PST-1 treated and untreated cells. **k**, **l** Area 2 selected in panels **e** and Supplementary Fig. 12a, respectively. **m** Calculated percentage of TPEF signal changes for the three time-windows before and after RPOC in Area 2 for cells treated with PST-1.

Although EB3 EGFP signals give direct visualization of microtubule dynamics, the weak intensity complicates the analysis. We used an indirect method to further validate the polymerization inhibition process by monitoring the LD dynamics. Mature LDs in cytosols are actively transported along microtubules using kinesin and dynein motors. Inhibition of the microtubule polymerization process can potentially stop the LD active transport. We use femtosecond SRS signals from $CH_2$ stretching to detect LDs and determine APXs for selective activation of PST-1 around LD pixels. Figure 8a, b show an example SRS image at 0 s and the APXs determined at 222 s. Two areas were selected for comparison. The cells were treated with 4 µM trans-PST-1 for 15 min. Then, 100 frames of SRS image are acquired before RPOC (Fig. 8c), from which high LD dynamics are detected as highlighted by arrows from 0–220 s. After turning on the RPOC, the dynamics of APX-associated LDs are significantly reduced from 222–442 s (Fig. 8d). In Area 2, APX-associated LDs also showed reduced dynamics after RPOC (Fig. 8e, f). However, the LDs that are not associated with APX (blue arrows in Fig. 8f) still have long-distance active transport during RPOC. Movie S10 shows changes in LD dynamics before and during RPOC. Figure 8g–i show the traced LD trajectories before RPOC, during RPOC, and only for LDs at the APXs, respectively. These results show that LDs at the APXs have much reduced long-distance trafficking compared to LDs that are not associated with APX. To quantify the changes in dynamics, we statistically analyzed the maximum displacement of LDs before RPOC (0–220 s), during RPOC

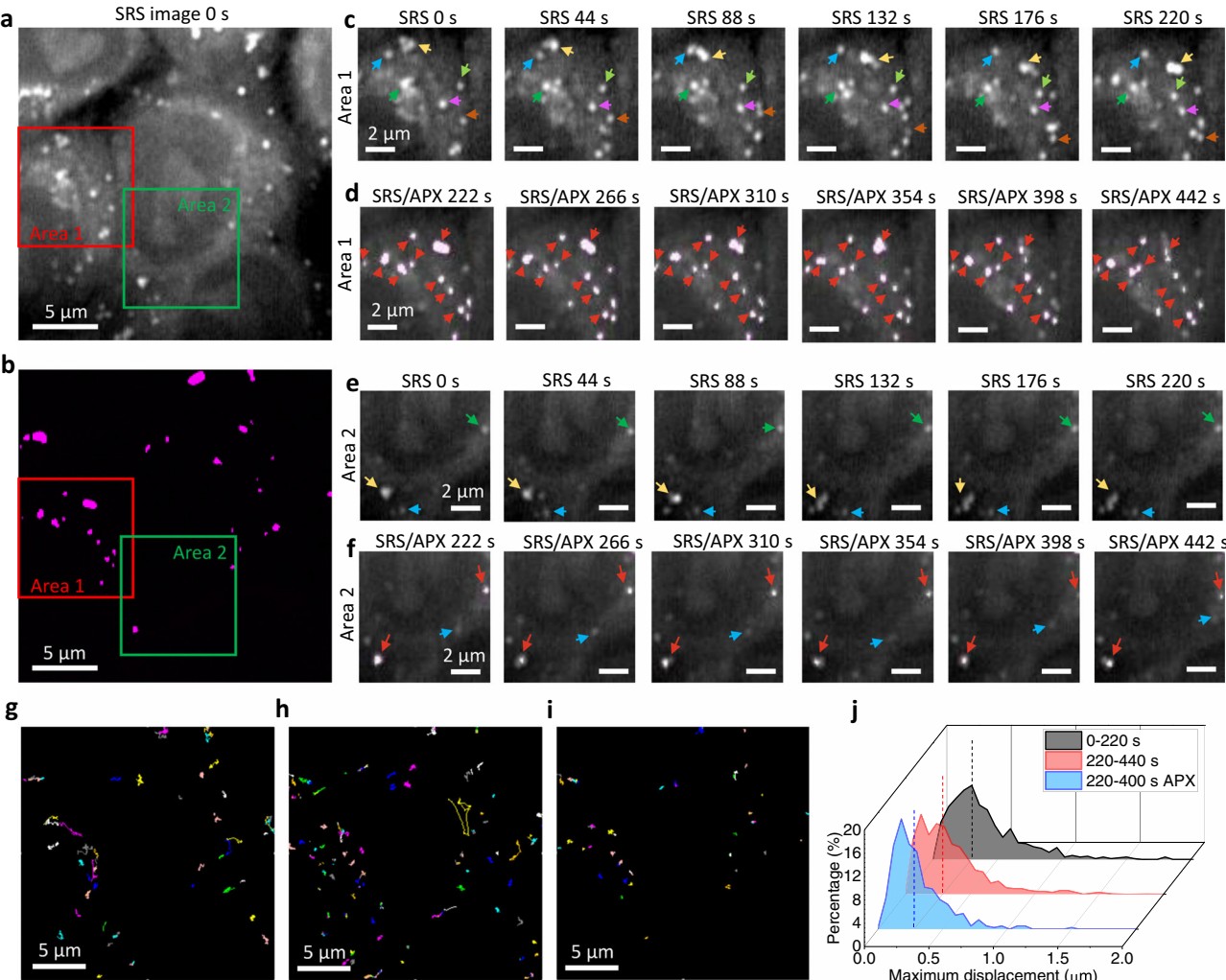

**Fig. 8 | Precision control of lipid droplet (LD) dynamics in live Kyoto HeLa cells.**
**a** A stimulated Raman scattering (SRS) image of Kyoto HeLa cells at time 0. **b** Active pixels (APXs) from the same field of view at time 222 s. **c** Time-lapse SRS images of LDs from time 0 s to 220 s for Area 1 in panel **a**. Arrows showing dynamics of LDs in cells. **d** Overlays of SRS images and APXs from time 222 s to 442 s for Area 1 in panel **a**. Arrows show reduced LD dynamics during RPOC. **e, f** Time-lapse images similar to panels **c** and **d** but from Area 2 in panel **a**. The blue arrows in panel **f** show the active transport of an LD not associated with APXs. The red arrows show LDs controlled by RPOC associated with APXs. **g** LD trajectories before RPOC. **h** LD trajectories during RPOC. **i** LD trajectories during RPOC from only the APX-associated LDs. **j** Histograms of LD maximum displacement before RPOC (gray), during RPOC (red), and during RPOC for only LDs at APXs (blue).

(220–440 s), and during RPOC for LDs associated with APXs using a previously reported method (Fig. 8j)[20,21]. A statistical reduction of maximum displacement can be seen during RPOC on LDs, and such a reduction is contributed by the LDs associated with APXs. The *cis*-PST-1 can spontaneously return to the *trans* form without blue light interaction[19]. We observed the reduction of LD dynamics during RPOC and the recovery of LD dynamics after RPOC (Supplementary Movie 11).

## Discussion

We demonstrate real-time precision opto-control of molecular activities and chemical processes triggered by optical signals from the molecules at submicron spatial precision. RPOC can perform active control of light-sensitive molecules and chemical reactions in living biological samples due to the fast response and automatic APX determination. Using an adjustable voltage reference for the comparator circuitry, we can easily tune conditions for APX determination for an unlimited range of samples and their respective signals of interest. The addition of digital logic functions to comparator circuits further enables multiple gating conditions for region-specific RPOC or higher chemical selectivity. Commercially available electronic

components are used to fabricate the comparator circuitry, as illustrated in Supplementary Figs. 1, 4. In this work, we majorly focused on demonstrating the RPOC capability using a photoswitchable molecule CMTE and the biological application of the precision control of PST-1 for the manipulation of microtubule-related cellular dynamics. RPOC can generally be applied to control newly developed photochromic vibrational probes[22,23], widely used photoswitchable fluorescent molecules[24,25], and light-sensitive chemical reactions[19,26–28] at high spatial and temporal accuracy. Photoremovable protecting groups can be used to convert small molecules into light-sensitive ones. When used with RPOC, such light-sensitive compounds can help to understand target locations in cells and subcellular compartments.

The continuous improvement of RPOC will lead to more opportunities in biophotonics and biological sciences. For example, further optimization of the control laser beam can improve the RPOC precision. Instead of using an expensive femtosecond laser, a more cost-effective and compact RPOC platform can be developed based on continuous-wave (CW) lasers. The CW-RPOC system would greatly reduce the system cost and is ideal for integration with commercial fluorescence microscopes. Programmable acousto-optic tunable filters would also allow for the selection of different laser beams

automatically for RPOC. Improvement in optics and electronics, such as using an electro-optic modulator and resonant mirrors would further improve the RPOC response time for high-speed laser scanning systems.

RPOC offers a way for biologists and chemists to control biomolecular behaviors and chemical reactions precisely and automatically in space and time without affecting unwanted targets. We believe RPOC will have important applications, when combined with photoactivatable molecules, for better control of enzyme activities, high accuracy-controlled release, high precision optogenetics, and improved precision treatment. Applying digital logic functions in RPOC with photoswitchable fluorescent molecules would also enable recording organelle interactions for live systems. Future research will focus on demonstrating the capabilities of RPOC in these applications.

## Methods

### The RPOC platform

A femtosecond laser source (InSight X3+, Spectra-Physics) is used for optical signal excitation and opto-control. The laser outputs two femtosecond pulse trains, one at a fixed wavelength of 1045 nm, and the other tunable from 690-1300 nm, both with ~100 femtosecond pulse width. For stimulated Raman scattering (SRS) microscopy, the 1045 nm output is used as the Stokes beam and the tunable output is used as the pump beam. A single 150 mm SF-57 glass rod (Lattice Electro-Optics) is placed in the Stokes beam and two 150 mm SF-57 glass rods are placed after combining the two laser beams for the chirping of the pump and Stokes pulses to 3.4 and 1.8 picosecond, respectively. The laser beams double-pass the two glass rods for additional chirping. Beam chirping is performed to allow for hyperspectral SRS imaging through spectral focusing. An acousto-optic modulator (AOM, M1205-P80L-0.5 with 532B-2 driver, Isomet) is used to modulate the Stokes beam at 2.7 MHz for SRS microscopy. Hyperspectral SRS image stacks are acquired by tuning the optical delay using a translational stage (X-LSM050A, Zaber Technologies) at 10 μm per step while collecting single-color images. The combined laser beams are directed to a 2D galvo scanner set (GVS002, Thorlabs) and then into an upright microscope frame (Olympus BX51). Either a 40x/0.8 NA (LUMPLFLN 40XW, Olympus) or a 60x/1.2 NA water immersion objective lens (UPLSAPO 60X, Olympus) is used to focus the laser beams onto the sample. Forward signals are collected using a 1.4 NA oil condenser. A 776 nm long-pass dichroic mirror (FF776-Di01-25 × 36, Semrock) is used to separate the TPEF signal from the input laser beams. The forward two-photon excitation fluorescence (TPEF) signals and the leaking of the control laser beam are detected by a photomultiplier tube (PMT) after being reflected by the long-pass dichroic mirror. A combination of filters (FF01-575/59-25 or FF01-451/106-25, Semrock; ET425lp, Chroma) is used to detect the fluorescence signal and the leaking of control laser beams. The SRS signals are detected after transmission of the dichroic mirror using a photodiode (S3994, Hamamatsu) paired with a lab-designed tuned amplifier with a center frequency of 2.7 MHz. A short-pass filter (980SP, Chroma Technology) is used to block the Stokes beam from entering the photodiode. A lock-in amplifier (HF2LI, Zurich Instruments) is used to demodulate signals for SRS imaging. The lock-in amplifier and the AOM for Stokes beam modulation are synchronized by a function generator (DG1022Z, Rigol). In the epi-direction, two PMTs are installed to collect fluorescence signals at selected wavelength windows using different filters. Chirping femtosecond laser pulses to picosecond is essential to perform hyperspectral SRS microscopy and better chemical selectivity. However, femtosecond excitation for TPEF and SRS is also available. The system is switchable between picosecond and femtosecond modalities via flip mirrors.

A portion of the tunable laser beam and the 1045 nm laser beam is frequency-doubled by Beta Barium Borate (BBO) crystals (EKSMA Optics) to generate visible wavelengths for opto-control. The crystals are mounted on rotational mounts to optimize the second harmonic generation efficiency at different wavelengths. Flip mirrors are used to select the desired control laser wavelength. The selected visible laser beam is sent to another AOM (M133-aQ80L-1.5 with 522B driver, Isomet) which is controlled by comparator circuit boxes. The optical signal voltage is compared with a preset condition to determine the output TTL voltage for AOM control. The profiles of the 0th and 1st order AOM outputs of the RPOC beam are collected using a CMOS camera (CinCam CMOS-1201) and processed with a Gaussian function fitting using commercial software (RayCi 64 bit). The SRS signal output is delivered from the lock-in amplifier, while the fluorescence signal output is delivered from an amplifier (PMT4V3, Advanced Research Instrument Corporation) connected after the PMT.

The design of the comparator circuit box 1 with a single intensity threshold is illustrated in Supplementary Fig. 1. Three operation modes of this circuit box include: AOM constantly on, AOM constantly off, and AOM control triggered by the signal-threshold comparison. Threshold voltage $V_T$ can be selected from either a manual tuning knob or a digital input with a range of 0-10 V and 0.01 V accuracy. The output TTL signal has a <0.7 V output as digital '0' and ~5 V output as digital '1'. The signal output and digital threshold output references are also available. The former can be used for image display during RPOC.

The design of the comparator circuit with digital logic is illustrated in Supplementary Fig. 4. Aside from the same functions as the comparator box 1, a TTL digital input is available allowing this comparator box to be used together with the comparator box 1 for digital logic selections. Digital logic functions can be selected by using jumpers on 3-pin jumper bars. Connections of AND, OR, NAND, and NOR functions are illustrated in Fig. 3d and Supplementary Fig. 10. The circuitry is fabricated with commercially available basic electronic components.

### Image acquisition and analysis

Images are saved as TXT files and processed using ImageJ for display. Pseudo-colors are used to represent different chemical compositions for SRS imaging and active pixels (APXs). Spectral or intensity profiles are plotted using Origin Pro. Particle trajectories are tracked using a particle tracker ImageJ plug-in[29]. The parameters to analyze the 100-frame time-lapse SRS image stack and the APX stack are: radius = 0, cutoff = 3, percentile = 0.5, link range = 1, displacement = 5. A single lipid droplet (LD) trajectory and the corresponding APX trajectory are plotted using the ImageJ particle tracker plug-in together with images for display. Merging different image channels, image subtractions, particle analyses, and intensity integrations are performed using ImageJ built-in functions. Hyperspectral SRS images are analyzed using a spectral phasor plug-in in ImageJ. Plots of chemical maps are pseudo-color-coded for display.

For statistical analysis of LD trajectories in Fig. 8, eight field-of-views containing 200 × 200 pixels are analyzed and all trajectories are summed to plot the histograms. The first 100 frames are collected before RPOC, while the next 100 frames are acquired during RPOC with APXs highlighted in magenta. APX images are used to trace trajectories of LDs only associated with APXs in Fig. 8. The maximum displacement of the trajectories is analyzed using lab-written MATLAB codes. Histogram plots are normalized by the number of trajectories that occupy >10 image frames. Example trajectories are plotted using the ImageJ particle tracker plug-in[29]. For Supplementary Movie 11, 300 frames of images are acquired, with the first 100 frames before RPOC, the next 100 frames during RPOC, and the last 100 frames after RPOC.

Quantitative analysis of TPEF signal changes for Fig. 7 is detailed in Supplementary Note 10.

### Chemical mixture preparation

Fluorescent microparticles including 1 μm green polystyrene (2104, 440/500 Ex/Em, Phosphorex, Inc.) and 1 μm orange polystyrene

(2205B, 525/580 Ex/Em, Phosphorex, Inc.) are mixed for fluorescence imaging in Fig. 2. Nonfluorescent polymer particles including 1 μm polystyrene (112, Phosphorex, Inc.) and 1 μm poly(methyl methacrylate) (MMA1000, Phosphorex, Inc.) are mixed for SRS imaging in Fig. 2. For Supplementary Fig. 8, the 1 μm green polystyrene and nonfluorescent polystyrene, as well as nicotinamide adenine dinucleotide hydrogen crystals (Cayman Chemical) are blended and dispensed on glass coverslips for imaging.

### Cell preparation
MIA PaCa-2 pancreatic cancer cells are purchased from ATCC. The HeLa Kyoto EB3-EGFP cells are purchased from Biohippo. Cells are cultured in Dulbecco's Modified Eagle Medium (DMEM, ATCC) with 10% fetal bovine serum (FBS, ATCC) and 1% penicillin/streptomycin (Thermofisher Scientific). The cells are seeded in glass-bottom dishes (MatTek Life Sciences) with 2 mL culture media and then incubated in a $CO_2$ incubator at 37 °C and 5% $CO_2$ concentration. Cells are grown to about 50% confluency and are directly used for live-cell imaging or fixed with 10% buffered formalin phosphate (Thermofisher Scientific) for imaging.

### Preparation of CMTE and control of CMTE in cells
The chemical $cis$−1,2-dicyano1,2-bis(2,4,5-trimethyl-3-thienyl)ethene (CMTE) is purchased from Sigma Aldrich and prepared in dimethyl sulfoxide (DMSO) at a concentration of 25 mM. MIA PaCa-2 cancer cells are treated with 3.2 μL of the CMTE stock solution for a final concentration of 40 μM. Cells are incubated with CMTE for 8–12 h before imaging. The combined pump and Stokes laser pulses can gradually switch the CMTE to the closed isomer **1b** with strong signals at 1510 cm$^{-1}$. We deploy RPOC to selectively convert CMTE at different locations of the sample to the open $cis$ isomer **1a**, as illustrated in Fig. 4b.

### Synthesis of PST-1 and cell treatment with PST-1
Procedures for the synthesis and characterization of PST-1 can be found in the Supplementary Note 9. The chemicals 3,4,5-trimethoxyaniline (T68209-10G), isopentyl nitrite (150495-100 ML), and pyrocatechol (C9510-100G) are purchased from MilliporeSigma. PST-1 is dissolved in DMSO at 2 mM and used as the stock solution. Before treatment, PST-1 is illuminated with a 532 nm laser (MSL-FN-532, Changchun New Industries Optoelectronics Tech. Co., Ltd.) for 5 s to convert the molecule to the $trans$ form. Cells are treated with PST-1 at a final concentration of 4 μM for 15 min before imaging and RPOC. To convert the PST-1 to the $cis$ form for UV-Vis measurement, a mounted LED (M405L4, Thorlabs) is used and the PST-1 is illuminated for 5 s.

### ER Tracker labeling of cells
MIA PaCa-2 cells are first seeded in glass-bottom dishes and cultured overnight to reach a 50–70% confluency. ER-Tracker is added to the culture medium with a 3 μM final concentration. The cells are cultured for 30 min at 37 °C and 5% $CO_2$ concentration before imaging. To generate sufficient TPEF signals, femtosecond laser pulses bypassing the chirping rods are directly used for signal excitation.

### Statistics and reproducibility
Statistical analysis in Fig. 8j is performed by quantitatively analyzing 8 image stacks randomly collected from the glass-bottom culture dish. No data are excluded from the analyses. Images in Figs. 2–6 are repeated at least 3 times with similar results. Images in Fig. 5 are repeated twice with similar results. Images in Fig. 8a–i are repeated 7 times to generate the statistical data in Fig. 8j. Images in the supplementary figures are repeated at least once.

### Reporting summary
Further information on research design is available in the Nature Research Reporting Summary linked to this article.

## Data availability
The data supporting the findings of this study are available within the paper and its supplementary information files. The image and analysis data generated in this study are deposited in the Figshare database. [https://figshare.com/s/c19c9929603955ad62c7]. Source data are provided with this paper.

## Code availability
The data acquisition code is lab-written based on LabVIEW. The data analysis code for LD dynamics is written using MATLAB. The codes can be accessed in Figshare [https://figshare.com/s/c19c9929603955ad62c7].

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

## Acknowledgements

This research is supported by the Purdue University startup fund, the NSF Major Research Instrumentation Award (2117616, C.Z.), the Center for Bioanalytic Metrology grant (an NSF Industry-University Cooperative Research Center under NSF IIP 1916991, C.Z.), and NIH R35 GM128570 (M.D.). The NIH P30 CA023168 is acknowledged for supporting shared NMR resources at Purdue Center for Cancer Research. We thank Jona-than Hood's lab for lending us the beam profiler and analysis software, and Garth Simpson's lab for lending us the autocorrelator for laser pulse width measurement.

## Author contributions

C.Z. designed the project and experiment and obtained funding for the research. C.Z. and M.G.C. performed the experiments, analyzed the results, and authored the paper. M.G.C. constructed the optical system for RPOC. G.A.G. helped prepare the cancer cells for imaging. Y.L. synthesized the PST-1 and performed the characterization. M.D. super-vised the synthesis of PST–1. J.A.A.-M helped in biological sample pre-paration. M.S.C. designed and fabricated the comparator circuits. G.E. designed and fabricated the tuned amplifier and photodetector for SRS signal detection.

## Competing interests

Authors declare that they have no competing interests.
