## [Peer Review File · Nature Communications]

Reviewers' Comments:

Reviewer #1:

Remarks to the Author:

The authors built a microscope-based instrument which allows them to demonstrate real-time precision opto-control of light responsive targets triggered by optical signals from the molecules at submicron spatial precision. The instrument itself is impressive and well engineered.

However, it is not clear at all what this instrument can be useful for. It can only control light-responsive targets, and the main technical advantage is to be able to trigger the light response it on-the-fly as compared to the standard method that do it after the imaging step. The Figures 4, 5, and 6 in the paper are mostly technical demonstrations without any chemical or biological insights. To reach the high bar of Nature Communications, their paper needs to demonstrate an application with clear and meaningful scientific results.

Reviewer #2:

Remarks to the Author:

In the work of Real-time precision opto-control of chemical processes in live cells, the authors developed a real-time precision opto-control (RPOC) technology triggered by optical signals obtained from a laser scanning system to detect chemical-specific optical responses from molecular targets during scanning. Benefits on the fast response and automatic active pixels determination, RPOC can perform active control of light-sensitive molecules and chemical reactions in living biological samples. This technology provided in-situ detecting methods with high spatial- and temporal resolution. The authors also demonstrated the applications of RPOC in dynamic living cell samples. The manuscript is written in a clear and concise style and most of the required information is given to substantiate all the claims by the authors. Overall, this study describes an interesting technology that can select and investigate some area/species via their optical response during imaging. However, the implementation of this technique is not attractive and possible applications of this technique seems limited. In this regard, the authors may have to use this technique to solve an important imaging or biological problem. Other than that, I have a few minor concerns in the following.

1. I wonder how the authors obtained the profile of the control laser beam after the AOM when the AOM is turned on and off, as shown in Figures 1D, and 1E? Why the first-order control light spot does not look like gauss distribution while AOM is turned on?
2. In the RPOC system, the authors show complicated excitation light paths. The tunable pulsed laser can obtain different wavelengths, However, as depicted in the optical configuration in Figure 1C, the control laser is frequency-doubled from excitation. Whether the wavelengths of the control laser can satisfy different optical sensitive targets?
3. Why broaden the pulse width chirping from 100 femtoseconds to few picosecond by pass through the glass rods? Furthermore, whether the dispersions of optical elements affect the signals of the laser scanning system and the spatial resolutions of localization of active pixels in RPOC?
4. A spelling mistake of "RPOC" in the first paragraph, page 2, and paragraph4, page6. "The spatial resolution of the SRS and TPEF modalities is measured to be 373 nm (fig. S2). The PROC control laser beam gives a spatial precision of 525 nm (fig. S3)."; PROC offers a way for biologists and chemists to control biomolecular behaviors and chemical reactions precisely and automatically in space and time without affecting unwanted targets.
5. The wavelength of the Stokes beam depicted in the main text is 1045 nm, while in Figures illustrations show 1040 nm (Figure 1C, Figure 4B). Please modify them to the correct number according to the actual condition.
6. "Parabola Fit of Sheet1 B" in Figure 4F, what does the Sheet1 B mean?

Reviewer #3:

Remarks to the Author:

Clark et al presents a novel high-speed opto-control with high spatial resolution. They achieve so by using a control laser beam that is has an optical feedback signal. This high-speed optical signal

goes through an engineered high-speed electronic system using multiple types of functions, in particular demonstrated with logic gates and thresholding. The feedback optical signals are showcased using nonlinear processes such as TPF and SRS contrast mechanisms. They demonstrate what such a machinery can achieve by performing realtime image thresholding, and they also claim to control chemical reactions that can be readout with high specificity using SRS signals. The paper is clearly written, although some times too technical in some obvious points.

While I found the idea interesting from an optical engineering perspective, the current version still misses what would this be useful for. For instance, the introduction they mention optogenetic has issues with spatial resolution, but that is not the main point of optogenetics: optogenetics is all about being able to find an optical way to control an electrical/chemical bio-signal (action potentials). That being said, the current manuscript feels mostly as an electronic engineering achievement, that I am afraid is not difficult to implement with off-the-shelf components and a modest electronic workshop. That is, for the moment, the current manuscript demonstrate a tool without giving it purpose, which I believe is of paramount importance for the readership of Nature Communications. In order to better understand whether this is indeed meaningful, I put up below some major remarks that the authors should consider for a revised manuscript

- 1) What are the fundamental mechanisms going on with control beam ON? What is it controlling/changing, in particular in the SRS process case that, in principle, cannot be controlled using such low powers? The case about the CMTE is clear, but the CH stretch is not intuitive. The same applies to the fluorescence results: what is the mechanism to shut down the fluorescence? Following this discussion, it would be nice to see that after the control mechanism, the images (or spectra) are not changed. That is, the control process is reversible.
- 2) The first example presented seems like a trivial thresholding one that I don't see what it can be useful for (related to fig. 2 results). One could achieve similar results with computational methods, so I wonder what is the advantage of such demonstration: could the method somehow reduce the shot noise of species that are controlled by the APX?
- 3) The examples using the digital logic controls are complex, but yet I don't see again what they could be useful for. Can the authors put up an explanation?
- 4) The only real applications shown are the results related to CMTE, but is there any other system that could benefit of such optical control? On the other hand, I found it nice to see that using APX, one could be less phototoxic compared to the conventional SRS imaging scheme.
- 5) I think the paper would benefit from a more in-depth discussion of the challenges related to the electronic implementation in the SI. What are the challenges in creating this electronics?

Minor points:

- 1) Can the author explain how the response time of the control beam can be much faster than the lock-in amplifier time constant? (assuming one uses the LIA demodulated output as feedback)
- 2) Check PBS cubes in Fig. 1 as they seem to be in wrong positions.

Reviewer #1 (Remarks to the Author):

The authors built a microscope-based instrument which allows them to demonstrate real-time precision opto-control of light responsive targets triggered by optical signals from the molecules at submicron spatial precision. The instrument itself is impressive and well engineered.

However, it is not clear at all what this instrument can be useful for. It can only control light-responsive targets, and the main technical advantage is to be able to trigger the light response it on-the-fly as compared to the standard method that do it after the imaging step. The Figures 4, 5, and 6 in the paper are mostly technical demonstrations without any chemical or biological insights. To reach the high bar of Nature Communications, their paper needs to demonstrate an application with clear and meaningful scientific results.

Reply: We thank the reviewer for saying that our instrument is impressive and well-engineered. To address the comment on what this instrument can do, we added new experiments and figures to demonstrate the control of microtubule polymerization and organelle dynamics in live cells (pages 6-7, figures 7 and 8, movies S6-S11). In this application, we synthesized a photoswitchable microtubule polymerization inhibitor PST-1. Using this molecule with RPOC, we demonstrated precision control of tubulin dynamics in selected areas of cells and the control of lipid droplet dynamics (trafficking) in live cells. Due to the inhibition of tubulin polymerization, RPOC can selectively reduce the active transport of lipid droplets at the APXs while not affecting others. This capability cannot be achieved with any existing technology. This application also highlights the potential to control activities of biomolecules at selected subcellular locations associated with specific molecular targets to understand site-specific molecular behaviors.

Reviewer #2 (Remarks to the Author):

In the work of Real-time precision opto-control of chemical processes in live cells, the authors developed a real-time precision opto-control (RPOC) technology triggered by optical signals obtained from a laser scanning system to detect chemical-specific optical responses from molecular targets during scanning. Benefits on the fast response and automatic active pixels determination, RPOC can perform active control of light-sensitive molecules and chemical reactions in living biological samples. This technology provided in-situ detecting methods with high spatial- and temporal resolution. The authors also demonstrated the applications of RPOC in dynamic living cell samples. The manuscript is written in a clear and concise style and most of the required information is given to substantiate all the claims by the authors. Overall, this study describes an interesting technology that can select and investigate some area/species via their optical response during imaging. However, the implementation of this technique is not attractive and possible applications of this technique seems limited. In this regard, the authors may

have to use this technique to solve an important imaging or biological problem. Other than that, I have a few minor concerns in the following.

Reply: We appreciate the reviewer for saying that our study describes an interesting technology. To address the comments on demonstrating an application, we added new experiments and a new section using RPOC to control tubulin polymerization at specification locations of cells and the dynamics of lipid droplets. (pages 6-7, figures 7 and 8, movies S6-S11) A photoswitchable tubulin inhibitor was synthesized which can be activated by blue light. Applying this molecule with RPOC, we showed site-specific inhibition of tubulin polymerization inhibition at subcellular locations. In addition, we showed control of lipid droplet trafficking due to the selective inhibition of tubulin polymerization only at the selected lipid droplet associated with APXs. Two new figures and 6 supporting movies are added to the manuscript to demonstrate this application. These results highlight the potential to use RPOC for site-specific control of molecular activities to understand location-specific enzyme functions and drug interactions. Photoswitchable inhibitors can be developed by adding photolabile protection groups to a wide variety of inhibitors for a better understanding of drug and biomolecular activities.

1. I wonder how the authors obtained the profile of the control laser beam after the AOM when the AOM is turned on and off, as shown in Figures 1D, and 1E? Why the first-order control light spot does not look like gauss distribution while AOM is turned on?

Reply: We used a camera to capture the light diffusion of the laser beam from a paper in the previous manuscript. To make a more valid analysis and measurement, we borrowed a beam profiler and measured the beam profiles at a larger distance from the crystal. New figures are used in Figs. 1D and 1E. The beams show Gaussian profiles.

2. In the RPOC system, the authors show complicated excitation light paths. The tunable pulsed laser can obtain different wavelengths, However, as depicted in the optical configuration in Figure 1C, the control laser is frequency-doubled from excitation. Whether the wavelengths of the control laser can satisfy different optical sensitive targets?

Reply: For our current setup, we opted to use the frequency-doubled output of the pump and Stokes beams because of the limitations of laser sources in the lab. Photo-activation of light-sensitive molecules does not require femtosecond lasers. CW lasers with broad wavelength selection can be coupled with the system for opto-control. Our current system has a tunable wavelength range of 350-520 nm and a separate 520 nm laser for opto-control. These wavelengths work for our photoswitchable molecules. If other wavelengths are needed, CW lasers of different wavelengths can be purchased and coupled to the beam path for RPOC opto-control.

3. Why broaden the pulse width chirping from 100 femtoseconds to few picosecond by pass through the glass rods? Furthermore, whether the dispersions of optical elements affect the signals of the laser scanning system and the spatial resolutions of localization of active pixels in RPOC?

Reply: The dispersion reduces the peak power but is essential for hyperspectral imaging. We broaden the pulse width from 100 fs to 1-3 ps to spectrally chirp the beams for hyperspectral SRS imaging of polymer particles Fig 2G-2I. This chirping also allows imaging of LDs and other chemical compositions in cells. Narrower spectral width gives better chemical selectivity for chemical detection. Our system can easily switch between fs and ps pulses using a few removable mirrors. For TPEF imaging in Fig 7, we bypassed the chirping rods and used femtosecond laser pulses directly for better signals. In addition, picosecond laser pulses reduce photodamage to live cells. We added a discussion in the method section. (Page 9)

4. A spelling mistake of “RPOC” in the first paragraph, page 2, and paragraph4, page6. “The spatial resolution of the SRS and TPEF modalities is measured to be 373 nm (fig. S2). The PROC control laser beam gives a spatial precision of 525 nm (fig. S3).”; PROC offers a way for biologists and chemists to control biomolecular behaviors and chemical reactions precisely and automatically in space and time without affecting unwanted targets.

Reply: Thank you. We have corrected this typo.

5. The wavelength of the Stokes beam depicted in the main text is 1045 nm, while in Figures illustrations show 1040 nm (Figure 1C, Figure 4B). Please modify them to the correct number according to the actual condition.

Reply: Our laser is centered at 1045 nm. We changed the content to be consistent.

6. “Parabola Fit of Sheet1 B” in Figure 4F, what does the Sheet1 B mean?

Reply: We have removed “of Sheet1 B” from the figure.

Reviewer #3 (Remarks to the Author):

Clark et al presents a novel high-speed opto-control with high spatial resolution. They achieve so by using a control laser beam that is has an optical feedback signal. This high-speed optical signal goes through an engineered high-speed electronic system using multiple types of functions, in particular demonstrated with logic gates and thresholding. The feedback optical signals are showcased using nonlinear processes such as TPF and SRS contrast mechanisms. They demonstrate what such a machinery can achieve by performing realtime image thresholding, and they also claim to control chemical reactions that can be readout with high specificity using SRS signals. The paper is clearly written, although some times too technical in some obvious points.

While I found the idea interesting from an optical engineering perspective, the current version still misses what would this be useful for. For instance, the introduction they mention optogenetic has issues with spatial resolution, but that is not the main point of optogenetics: optogenetics is all about being able to find an optical way to control an electrical/chemical bio-signal (action potentials). That being said, the current manuscript

feels mostly as an electronic engineering achievement, that I am afraid is not difficult to implement with off-the-shelf components and a modest electronic workshop. That is, for the moment, the current manuscript demonstrate a tool without giving it purpose, which I believe is of paramount importance for the readership of Nature Communications. In order to better understand whether this is indeed meaningful, I put up below some major remarks that the authors should consider for a revised manuscript

Reply: We thank the reviewer for stating our idea interesting. To address the comment “the current manuscript demonstrates a tool without giving it purpose”, we added new experiments to highlight the capabilities of RPOC and reiterate the potential of this new approach. (pages 6-7, figures 7 and 8, movies S6-S11). In the new experiments, we synthesized a photoswitchable tubulin polymerization inhibitor which can be activated by blue light. Using RPOC, we demonstrated selective activation of this inhibitor in a subcellular position by observing the tubulin polymerization dynamics and lipid droplet trafficking changes. The selective inhibition of tubulin polymerization reduces the active transport of lipid droplets in live cells associated with active pixels while not affecting the lipid droplet dynamics outside the active pixels. This application indicates that we can selectively control the molecular behaviors at only the wanted locations by pairing RPOC with photoswitchable compounds. Photoswitchable inhibitors can be developed by adding photolabile protection groups to the original inhibitor. Using with RPOC, these photosensitive compounds would allow a better understanding of drug-target interactions and control of enzyme activities only at desired locations of cells. This capability is currently unavailable in biological science.

1) What are the fundamental mechanisms going on with control beam ON? What is it controlling/changing, in particular in the SRS process case that, in principle, cannot be controlled using such low powers? The case about the CMTE is clear, but the CH stretch is not intuitive. The same applies to the fluorescence results: what is the mechanism to shut down the fluorescence? Following this discussion, it would be nice to see that after the control mechanism, the images (or spectra) are not changed. That is, the control process is reversible.

Reply: When the control beam is ON, it triggers chemical reactions, e.g. CMTE and PST-1 molecules. The CH stretching and fluorescence signals will not be affected since the laser beams used for control have much lower power (10 μ W) compared to the imaging laser beams (1-10 mW). RPOC will not affect CH and fluorescence signals but use these signals to determine where to turn on the control laser beam. If CH vibration is used in SRS, the control beam is mostly turned on for lipid droplets; if the fluorescence signal is used, the control beam is turned on at the fluorescence pixels, e.g. ER after labeling. For Fig 6, the SRS CH signals are collected after RPOC. The CH signal intensities are not changed. Another example is shown in the newly added movie S11, from which the CH SRS images before, during, and after RPOC were acquired and the control beam is not affecting the SRS signals. The low power control beam also contributes to negligible photobleaching compared to the femtosecond laser pulses for fluorescence signal excitation.

2) The first example presented seems like a trivial thresholding one that I don't see what it can be useful for (related to fig. 2 results). One could achieve similar results with

computational methods, so I wonder what is the advantage of such demonstration: could the method somehow reduce the shot noise of species that are controlled by the APX?

Reply: The goal of RPOC is not to reduce shot noise but to control biomolecular behaviors at selected pixels. The goal of Fig 2 is to show how to determine the active pixels (where the control laser is turned on) using a intensity thresholds. This is the simplest method for decision making based on the chemical compositions in an image. More versatile active pixel determinations are shown in Fig 3 and 5. The control of CMTE in Fig 4 and LD dynamics in Fig 8 uses the intensity thresholds as decision making criteria.

3) The examples using the digital logic controls are complex, but yet I don't see again what they could be useful for. Can the authors put up an explanation?

Reply: The digital logic functions allow to (1) select intensity ranges for active pixel determination instead of a single threshold, and (2) select active pixels associated with two or more detection channels. One example is in Fig 6. RPOC can selectively convert CMTE to the open form only at the LDs that are on the ER, not outside the ER. Such a feature makes RPOC capability to control molecular activities with more spatial and chemical specificity. Another example is Fig 5, which shows using intensity ranges to control the state of CMTE at the edges of the aggregates.

4) The only real applications shown are the results related to CMTE, but is there any other system that could benefit of such optical control? On the other hand, I found it nice to see that using APX, one could be less phototoxic compared to the conventional SRS imaging scheme.

Reply: Please see the reply to the general comments. We added the application of RPOC with a photoswitchable tubulin polymerization inhibitor to precisely control the tubulin dynamics as well as lipid droplet trafficking in live cells. For the second comment, APX will not make the system less phototoxic compared to the conventional SRS imaging scheme. Here, RPOC uses SRS signals to turn on another very weak opto-control beam (10 microwatts) for molecular control. It will not add noticeable phototoxicity to the SRS system due to the low opto-control laser power.

5) I think the paper would benefit from a more in-depth discussion of the challenges related to the electronic implementation in the SI. What are the challenges in creating this electronics?

Reply: We use common electronics as illustrated in Figures S1 and S4. The circuit designs are included in these figures. We don't think there are any challenges in creating these comparator circuits and we believe any electronic shop can fabricate these circuits or modified versions. We added a sentence to state this in the discussion and also in the methods. (page 8 and page 10)

Minor points:

1) Can the author explain how the response time of the control beam can be much faster than the lock-in amplifier time constant? (assuming one uses the LIA demodulated output as feedback)

Reply: We thank the reviewer for this comment. The response time for the fluorescence detection channel is at the level of 50 ns. However, for the SRS detection, it is mostly determined by the time constant of lock-in demodulation for SRS signals, which is 7 microseconds in our case. Reducing the time constant can increase the response time of the SRS-triggered RPOC. This still allows activation of the same APX as the signal pixel and possibly affects the next scanning pixel on the image. In our oversampling condition, such an impact is within 90 nm. To clarify this, we added a discussion in the main content (page 3) and in the SI (The RPOC response time).

2) Check PBS cubes in Fig. 1 as they seem to be in wrong positions.

Reply: Thank you and we fixed this error.

Reviewers' Comments:

Reviewer #1:

Remarks to the Author:

The newly added experiments demonstrated the control of microtubule polymerization and organelle dynamics in live cells. This application would be hard to achieve with other existing technology. Given this new demonstration, I now support its publication in NATURE COMM.

Reviewer #2:

Remarks to the Author:

In the first review I requested a meaningful biological application by using the RPOC technology. The authors performed additional experiments and added a new section in the revised manuscript to demonstrate the usefulness of the proposed technology. Specifically, the authors showed site-specific inhibition of tubulin polymerization inhibition at subcellular locations and control of lipid droplet trafficking due to the selective inhibition of tubulin polymerization only at the selected lipid droplet associated with APXs. I think these biological studies are interesting and confirm that RPOC could be a useful method in subcellular manipulation and control of cellular events. Therefore, I recommend publication of this paper on Nature Communications.

Reviewer #3:

Remarks to the Author:

In this new version, the authors have addressed the critical issue of the bio-applicability of their technique. They added two new results on polymerisation control of microtubule, which has a mechanism established in the literature (Ref. 21). In principle, they show an application in biological samples, but it is nowhere clear of why one needs such an insane high-resolution control of chemical processes. In the reply they quote "Using with RPOC, these photosensitive compounds would allow a better understanding of drug-target interactions and control of enzyme activities only at desired locations of cells.": this is the type of sentence, together with a few examples, that I would expect in a good introduction of a paper with such a tool (not what they currently use in 1st paragraph which is too vague and unclear of the need to get such a high-resolution). At the moment, for instance, it is not clear to the reader why using this tool to control the polymerisation of microtubules is needed: I had to read Ref. 21 to understand its relation to drug-target interactions, but clearly only cell-level resolution is needed in Ref. 21 (as it is related to chemotherapy). I believe the authors have at least one result with relevance for biology, but there is a real need of re-writing the introduction/discussion to address why this technology is useful. Some major comments below:

- 1) I found the result of figure 7 not convincing (and with some scanning artefacts showing as stripes on Fig. 7M/N): with so much noise, can the authors show at least some statistics to convince the reader?
- 2) Conversely, fig. 8 is more convincing, judging from the statistical analysis of fig. 8J.

Reply to reviewers' comments

Reviewer #1 (Remarks to the Author):

The newly added experiments demonstrated the control of microtubule polymerization and organelle dynamics in live cells. This application would be hard to achieve with other existing technology. Given this new demonstration, I now support its publication in NATURE COMM.

Reply: We thank the reviewer for supporting the publication of this manuscript.

Reviewer #2 (Remarks to the Author):

In the first review I requested a meaningful biological application by using the RPOC technology. The authors performed additional experiments and added a new section in the revised manuscript to demonstrate the usefulness of the proposed technology. Specifically, the authors showed site-specific inhibition of tubulin polymerization inhibition at subcellular locations and control of lipid droplet trafficking due to the selective inhibition of tubulin polymerization only at the selected lipid droplet associated with APXs. I think these biological studies are interesting and confirm that RPOC could be a useful method in subcellular manipulation and control of cellular events. Therefore, I recommend publication of this paper on Nature Communications.

Reply: We thank the reviewer for recommending the publication of our paper in Nature Communications.

Reviewer #3 (Remarks to the Author):

In this new version, the authors have addressed the critical issue of the bio-applicability of their technique. They added two new results on polymerisation control of microtubule, which has a mechanism established in the literature (Ref. 21). In principle, they show an application in biological samples, but it is nowhere clear of why one needs such an insane high-resolution control of chemical processes. In the reply they quote “Using with RPOC, these photosensitive compounds would allow a better understanding of drug-target interactions

and control of enzyme activities only at desired locations of cells.”: this is the type of sentence, together with a few examples, that I would expect in a good introduction of a paper with such a tool (not what they currently use in 1st paragraph which is too vague and unclear of the need to get such a high-resolution). At the moment, for instance, it is not clear to the reader why using this tool to control the polymerisation of microtubules is needed: I had to read Ref. 21 to understand its relation to drug-target interactions, but clearly only cell-level resolution is needed in Ref. 21 (as it is related to chemotherapy). I believe the authors have at least one result with relevance for biology, but there is a real

need of re-writing the introduction/discussion to address why this technology is useful.
Some major comments below:

1) I found the result of figure 7 not convincing (and with some scanning artefacts showing as stripes on Fig. 7M/N): with so much noise, can the authors show at least some statistics to convince the reader?

2) Conversely, fig. 8 is more convincing, judging from the statistical analysis of fig. 8J.

Reply: According to the reviewer's suggestions, we made two major changes:

1. Rewrote the introduction, paragraph 1. (Page 2)
2. Modified Figure 7 by adding quantitative analysis and added a few supplementary figures. An explanation and discussion on the quantitative analysis results are also added to the manuscript and the supplementary information. (Page 7)

We want to thank this reviewer for detailed suggestions and critical comments to help improve our manuscript.

Reviewers' Comments:

Reviewer #3:

Remarks to the Author:

The authors have addressed all my previous comments, in particular providing more convincing statistical analysis on Fig. 7, and explaining better why such high-resolution optical control could be useful for. Therefore, I suggest publication in Nature Communications

Reply to reviewers' comments

Reviewer #3 (Remarks to the Author):

The authors have addressed all my previous comments, in particular providing more convincing statistical analysis on Fig. 7, and explaining better why such high-resolution optical control could be useful for. Therefore, I suggest publication in Nature Communications

Reply: We thank the reviewer for suggesting the publication of this manuscript.